# “The Difference between Plan b and ella®? They’re Basically the Same Thing”: Results from a Mystery Client Study

**DOI:** 10.3390/pharmacy8020077

**Published:** 2020-05-01

**Authors:** Guneet Kaur, Tiana Fontanilla, Holly Bullock, Mary Tschann

**Affiliations:** 1Division of Narrative Medicine, Columbia University, New York, NY 10027, USA; 2Department of Obstetrics, Gynecology & Women’s Health, University of Hawaiʻi at Manoa, Honolulu, HI 96826, USA; tianamf@hawaii.edu (T.F.); marytschann@gmail.com (M.T.); 3Department of Obstetrics and Gynecology, University of Arizona, Tucson, AZ 85724-5078, USA; hollybullock@obgyn.arizona.edu

**Keywords:** ulipristal acetate, emergency contraception, pharmacies, United States, Hawaiʻi

## Abstract

Pharmacy staff can serve an important role educating patients about emergency contraceptive pills (ECP), particularly ulipristal acetate (UPA), which requires a prescription. We conducted a secondary analysis of a previously completed mystery client study, assessing accuracy of information provided by pharmacy staffers to patients inquiring by telephone about filling a prescription for UPA. From the period December 2013 to July 2014, researchers used a mystery client methodology, contacting 198 retail pharmacies in Hawaiʻi. Researchers posed as patients or providers attempting to fill a prescription for UPA. During the course of the call, they asked about differences between UPA and levonorgestrel ECPs. Nearly half of all pharmacy staffers were unfamiliar with UPA. The majority of responses describing differences between the medications were incorrect or misleading, such as responses implying that UPA is an abortifacient. Lack of familiarity and incorrect information provided by pharmacy staffers may act as additional barriers in patient access to UPA. Health practitioners prescribing UPA should ensure patients receive evidence-based counseling at the time of prescription, while efforts should also be made to improve pharmacy staff familiarity with emergency contraceptive options.

## 1. Introduction

Ulipristal acetate (UPA), a second-generation selective progesterone receptor modulator sold in the US as ella® (30 mg), was approved in 2010 by the US Food and Drug Administration (FDA) for use as an emergency contraceptive pill (ECP). UPA is recommended for use within 120 hours of unprotected sexual intercourse and is more effective at preventing ovulation and pregnancy when compared to levonorgestrel ECPs (LNG ECPs) at one, three, and five days [1]. LNG ECPs include Plan B One Step®, Take Action®, My Way®, AfterPill®, and more.

Access to UPA through retail pharmacies is limited when compared to LNG ECPs as few pharmacies carry the medication [2]. A 2018 mystery client study of over 500 pharmacies across 10 large cities found that less than 10% of pharmacies were able to immediately fill a UPA prescription [3]. Clinicians are also less familiar with UPA than LNG ECPs. A 2016 survey assessing knowledge of ECPs among health care providers demonstrated that 95% of respondents had heard of LNG and 81% had provided it. However, outside of reproductive health specialties, only 18% of internists and 14% of emergency medicine providers had heard of UPA and 4% had provided it [4]. Requirement of a prescription and lack of prescriber knowledge about UPA are contributors to these disparities between LNG and UPA access [2]. 

Misinformation given to patients when trying to fill prescriptions may also contribute to access barriers. In a 2011 mystery client study in Iowa, some pharmacists supplied misinformation about LNG ECPs, indicating that it is an abortifacient, is associated with birth defects, and is unsafe for adolescents [5]. A 2013 nationwide mystery client survey with callers posing as 17-year-old women seeking LNG ECPs also exposed additional barriers to access, with pharmacy staff incorrectly indicating that the medication required a prescription, could not be bought by minors or by male customers, stating that parental notification was required, and that the purchaser needed to present valid government-issued identification [6].

Currently, there is no published data describing the information provided by pharmacy staff regarding UPA. We conducted a secondary analysis of a previously published mystery client telephone survey that assessed statewide availability of UPA and LNG ECPs in Hawai’i that included verbatim responses from pharmacy personnel regarding the differences between UPA and LNG ECPs. Our outcomes were to categorize the type of information voluntarily provided over the phone and to assess the accuracy of these responses.

## 2. Materials and Methods

We performed a secondary analysis of an observational population-based dataset assessing statewide availability of ECPs in retail pharmacies [2]. Two online phone books were cross-referenced with pharmacy chain websites to generate a list of all 198 unique retail pharmacies in Hawaiʻi (Figure 1). Pharmacies were excluded if the phone number was found to be disconnected or incorrect or if research staff was unable to reach the pharmacy after three attempted calls. Additional exclusions included pharmacies unwilling to provide information over the telephone and incomplete data collection at the time of call. Additionally, pharmacies that reported not carrying ECP, did not provide information to patients over the phone, and specialty pharmacies were excluded. This study was reviewed by University of Hawaiʻi Institutional Review Board which deemed it to be “not human subjects research”.

Trained research assistants posing as 18-year-old patients attempting to fill a UPA prescription made calls to pharmacies utilizing a semi-structured questionnaire asking about the pharmacy’s ability to fill a prescription for UPA. They also asked about the availability of other types of ECPs and concluded with an open-ended question about the difference between UPA and LNG ECPs A physician member of the research team also made calls to pharmacies utilizing a similar semi-structured questionnaire, attempting to identify pharmacies where their patient could fill a UPA prescription, if additional LNG ECPs were available, but only asked the open-ended question about the difference between UPA and LNG ECPs if the pharmacy carried any ECPs. After pilot testing amongst departmental faculty and beta testing with 10 pharmacies in North Carolina, having the physician caller solicit more information about medications not on formulary was deemed inauthentic to reality, whereas for the patient caller soliciting information was deemed authentic to a real world patient interaction (Figure 2). 

The callers documented the pharmacy personnel responses to all questions, including additional remarks made throughout the call. Calls were made Monday through Saturday, 8 a.m. through 8 p.m., from the period December 2013 to July 2014. Three unique attempts were made to each pharmacy by the physician and patient mystery clients, varying the time of day and the day of the week. Calls were staggered by 4 weeks between the physician and patient mystery clients. 

In order to ensure accurate replication of clinical reality, researchers did not specifically ask to speak to a pharmacist. Rather, they posed questions to whichever staff member answered the phone. To minimize potential bias, mystery clients did not inform pharmacy staffers that calls were part of a research study. 

For this secondary analysis, we analyzed verbatim answers to the question “Is there a difference between ella® and Plan B?” (Figure 2) and any additional related responses that were documented by the researchers during the course of the semi-structured interview. Two authors (GK, MT) separately conducted content analysis, which allows for the distillation of responses into overarching categories and response counting. Discrepancies in coding were discussed to reach consensus in classification of category. Each author classified answers as correct or incorrect information using FDA Fact sheets [7,8]. Discrepancies were again discussed to reach agreement.

Answers were grouped into categories of response for both medications: timing of administration, mechanism of action, over the counter availability (and related restrictions) efficacy, and drug type. Correct answers for each medication by category can be found in Table 1.

## 3. Results

Of the 358 total mystery client calls, 192 were patient calls and 166 were physician calls with a total of 198 pharmacies contacted. We identified patterns of incorrect information provided among the responses. We found no meaningful differences between the frequency of correct and incorrect responses given to patient or physician mystery clients.

Nearly half of all pharmacy staffers (39%, 141/358) expressed a lack of familiarity with UPA, expressing that they had never heard of UPA, did not know much about it, and were generally more familiar with LNG ECPs (Table 2). We found that 61% (217/358) of pharmacy personnel indicated familiarity with both LNG and UPA, while 30% (107/358) of respondents indicated that they were completely unsure of the differences between the two drugs, and 9% (34/358) of respondents indicated that they were accessing online information during the call so as to offer accurate information to the mystery caller. In 40% (56/141) of the calls among those who expressed a lack of familiarity with UPA, the initial respondent transferred the call to a pharmacist or more knowledgeable colleague. 

When asked about differences between LNG and UPA ECPs, respondents most frequently stated that UPA and LNG were different drugs. Of the 55% (197/358) of respondents who addressed this difference, 29% (58/197) correctly identified UPA as a selective progesterone reuptake modulator and LNG as a hormone-based pill, but incorrect information was provided by 71% (139/197) of respondents. Two illustrative examples of correct and incorrect statements about drug types are below:

Correct:
“ella® is a selective progesterone modulator—blocks progesterone receptors, inhibits ovulation and possibly prevents implantation. Plan B is a hormone pill.”

Incorrect:
“They are both emergency contraceptives, high dose hormones. They both work the same way.”

Nearly 39% (139/358) of respondents stated timing of drug administration as a difference between the two medications. Of these, 81% (114/139) incorrectly identified the differences in timing and efficacy of UPA and LNG. Examples of some of the incorrect statements provided are below:

Incorrect:
“ella® can be used up to four days after intercourse, but Plan B is within 72 hours.”
“ella® is taken 72 hours after intercourse. Plan B is taken the day after.”

Approximately 34% (122/358) of respondents identified mechanism of action as being different between UPA and LNG. Of these, 34% (42/122) correctly identified the mechanism of action for both UPA and LNG, while the remaining statements were incorrect, including:

Incorrect:
“Plan B is the hormone—you overload your body. [It’s] like taking five birth control pills. ella® helps the person push [the pregnancy] out.”
“ella® is the abortion pill, you take it if you are already pregnant.”

Approximately 11% (41/358) of respondents mentioned differences in prescription requirement. Nearly three quarters of these respondents (71%, 29/41) correctly stated that UPA requires a prescription and LNG does not. 

Over ten percent of respondents (12%, 44/358) mentioned differences in efficacy rates between UPA and LNG. A quarter of these responses (25%, 11/44) correctly described UPA as being more effective for patients with higher BMI. Statements regarding efficacy included:

Correct:
“ella® is just a different medication. ella® is more effective [than LNG], but not available.”
“There was a study on the Plan B regarding the patient weight, and ella® was better if the patient was bigger.”

## 4. Discussion

We examined the accuracy of information provided by retail pharmacy personnel when asked to describe differences between UPA and LNG during a call inquiring about a prescription for UPA. Callers seeking such information from retail pharmacies in Hawaiʻi received accurate information from a small percentage of respondents. Lack of familiarity with UPA was a common theme encountered. This could be linked to the limited availability of UPA in pharmacies, as only 3% of retail pharmacies reported same day availability and 25% reported the ability to order UPA in the primary analysis of this data [2]. To date, there has been no published literature on information that callers receive over the phone from pharmacies regarding UPA, but there is a large body of literature citing misinformation from pharmacy staff to callers for LNG ECPs. As the majority of patients’ access ECPs from chain pharmacies or “big box” stores, we sought to address this gap in literature [9]. 

Our secondary analysis of this data has some meaningful limitations. First, each caller spoke with whichever pharmacy staffer answered the phones. In many cases, it was unclear whom mystery callers spoke with as asking about the pharmacy personnel’s role was not part of the semi-structured questionnaire. However, this omission was intentional, in an effort to preserve the authenticity of the call. 

Additionally, throughout our analysis we marked responses dichotomously as correct or incorrect; however, some responses may have been mostly or partially correct or incorrect. Two researchers verified the accuracy of each statement with the FDA fact sheet before it was categorized. Furthermore, mystery client callers did not probe pharmacy staffers for specific information, as we did not establish categories of inquiry about differences between UPA and LNG ECPs as part of the questionnaire. Pharmacy staffers, therefore, offered varying levels of information when asked to identify differences between UPA and LNG ECPs, as would occur naturally during a call to a pharmacy. As a result, we have variable denominators for each topic addressed, which makes comparison challenging. These denominators only reflect the number of pharmacy personnel who volunteered information about a given topic. It is possible that some pharmacy personnel were knowledgeable of additional key information about ECPs but just did not mention it during the call. Therefore, our analysis may not accurately reflect the true proportion of correct responses had pharmacy staffers been probed to address each topic. Additionally, because the callers were specifically asking about filling a prescription, this could have impacted the frequency with which pharmacy staffers offered this information. 

Additionally, in our data analysis, we also faced challenges in navigating the nuances of FDA fact and evidence-based medicine, particularly with identifying parameters of answers deemed "correct." There are two major elements of the FDA label for LNG EC that many researchers consider to be incorrect: the timing and the mechanism of action. Some data suggest that LNG EC can be effective for up to 96 or 120 hours, so many clinicians consider the 72-hour limit to be obsolete [10]. In addition, the label states that prevention of implantation is a possible mechanism of LNG EC; this has not been updated even though the best available evidence suggests that LNG EC works only before ovulation [11]. As such, it is difficult to determine what is correct when the best available research conflicts with the FDA label. For the purposes of our data analysis, we considered a statement correct if it agrees with the FDA label, though some of the language on the label is out of date.

Finally, it is important to acknowledge that these data were collected a few years ago. Because a new company took over the distribution of ella® in 2014, more recent data are needed to more accurately assess if changes in distribution have impacted pharmacy personnel’s familiarity with the medication. 

Lack of familiarity with UPA and incorrect information provided to patients and providers are barriers to timely access to and correct use of ECPs. Pharmacists frequently gave incorrect information about characteristics of ECPs that are most critical for patients to understand when using the medication: timing of drug administration, efficacy, and mechanism of action. Patients need to know how to use a medication and to understand if an alternative medication, such as LNG ECPs, works as well as the medication they were prescribed. Incorrect information, stating that UPA is an abortifacient, may deter some patients from using it or may encourage patients to use it after already confirming a pregnancy. The proliferation of incorrect information by trusted clinical professionals, including pharmacy personnel, has meaningful and reverberating impacts on a patient’s experience with a medication. Clinicians who prescribe UPA should be aware of the potential for patients to encounter incomplete or incorrect information from pharmacy personnel and should proactively provide patients with thorough counseling about the medication and its appropriate use at the time of prescribing. Additionally, partnerships between leaders in a pharmacy and family planning should be pursued in order to improve pharmacist and pharmacy staff capacity to engage with patients with accurate information about emergency contraception. 

## Figures and Tables

**Figure 1 pharmacy-08-00077-f001:**
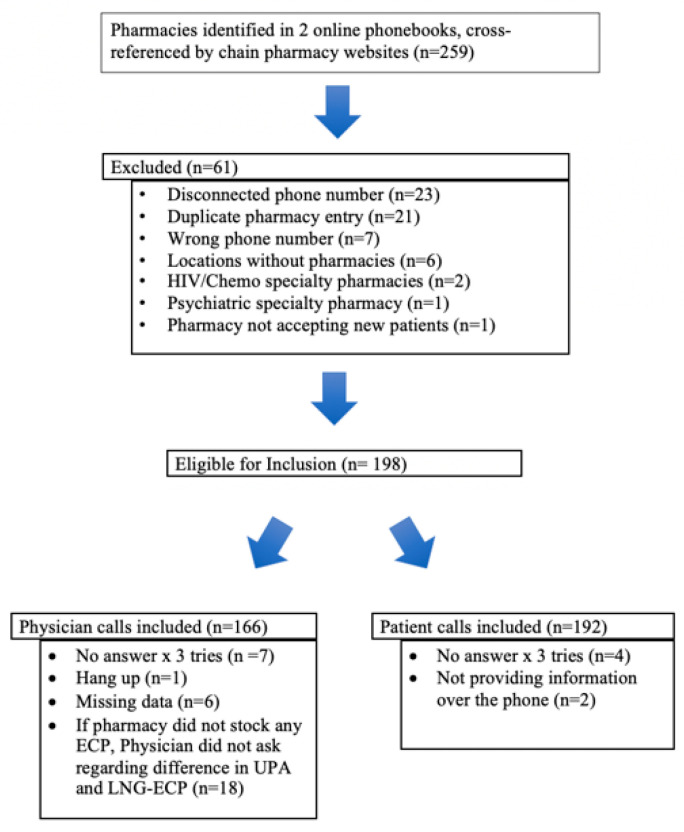
Inclusion and exclusion criteria for pharmacies.

**Figure 2 pharmacy-08-00077-f002:**
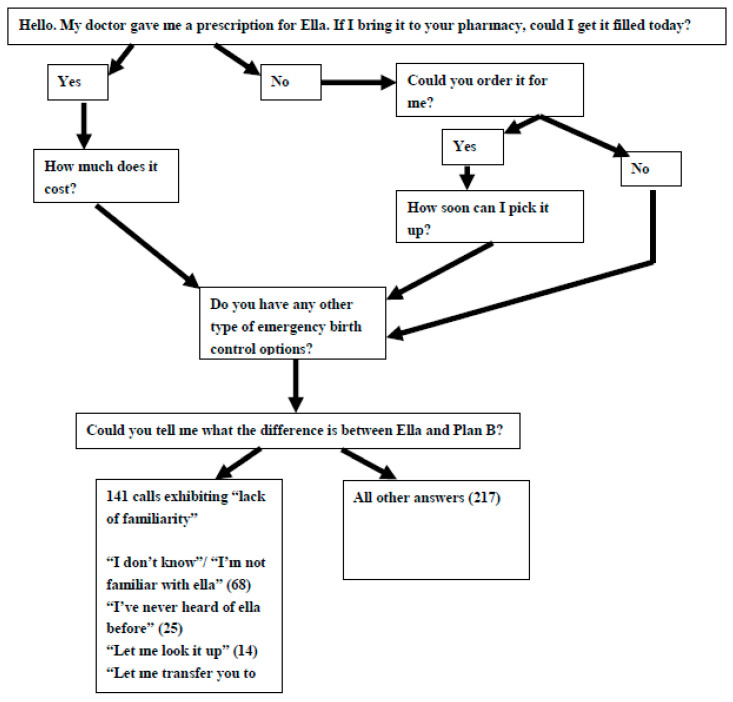
Sample questionnaire for mystery client call.

**Table 1 pharmacy-08-00077-t001:** A breakdown of answers accepted as correct for ulipristal acetate (UPA) and levonorgestrel (LNG). Our content analysis generated five categories of answers differentiating UPA from LNG: timing of drug administration, prescription requirements, efficacy, mechanism of action, and drug type.

	Ulipristal Acetate (UPA)	Levonorgestrel (LNG)
Timing of Drug Administration	Answers were deemed correct if they mentioned the 120-hour time frame.	Answers were deemed correct if they mentioned the 72-hour time frame.
Prescription Requirement	Answers were deemed correct if they noted the prescription requirement.	Answers were deemed correct if they noted the over-the-counter availability.
Drug Type	Answers were deemed correct if they noted that UPA is a selective progesterone receptor modulator.	Answers were deemed correct if they noted that LNG is a progestin-only contraceptive.
Mechanism of Action	Answers were deemed correct if they noted that UPA works by preventing or delaying ovulation.	Answers were deemed correct if they noted that LNG works by preventing or delaying ovulation.
Efficacy	Answers were deemed correct if they noted that UPA tends to be more effective for folks of overweight and obsese BMI classifications.	Answers were deemed correct if they noted that LNG tends to be more effective for folks of low/normal BMI classifications.

**Table 2 pharmacy-08-00077-t002:** Frequencies of correctly identified differences between UPA and LNG. Results are shown as overall frequency of each category addressed as well as frequencies of correct responses for each category (all n = 358).

Category	Frequency of Each Category Addressed		Frequency of Correct Responses	
Lack of Familiarity with UPA	39%	(141/358)	--	
Timing of Drug Administration	39%	(139/358)	18%	(25/139)
Prescription Requirement	12%	(41/358)	71%	(29/41)
Drug Type	55%	(197/358)	29%	(58/197)
Efficacy	12%	(44/358)	25%	(11/44)

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
