# Peer review of "“The Difference between Plan b and ella®? They’re Basically the Same Thing”: Results from a Mystery Client Study"

_pharmacy, 2020, doi:10.3390/pharmacy8020077_

Round 1
Reviewer 1 Report
This is an interesting mystery caller study regarding pharmacists' knowledge of the difference between UPA and LNG ECPs.
Specific Edits:
Intro
- UPA--it would be helpful to mention that UPA's effectiveness doesn't diminish over time (as ECP's does).
- Line 51-suggest stating "government-issued identification" instead of "ID"
- The authors make the argument that overall knowledge of UPA is low (amongst health care providers) and availability--can they clarify their hypothesis for this study to examine pharmacy staff knowledge?
Methods
- The table of "correct" answers lists LNG as being effective for 72hr time-frame, can the authors clarify if the 5-day window that is evidence-based was also acceptable?
- It would be helpful if authors could help map the outcomes reported in Results and Table 2 onto the questions that were asked during the script and how answers were characterized. Table 1 helps talk about what was considered "correct", but then they report "lack of familiarity" and therefore it isn't clear how/why calls were characterized that way. Table 1 and Table 2 could list categories (Efficacy, Mechanism, etc) in the same order
Results
- Can the authors clarify how there were 198 pharmacies called-but only 192 were mystery caller patients and 166 were physicians? Was every pharmacy not called by both the patient and physician? If not, can they clarify in methods?
- The outcome of "lack of familiarity with UPA"--was that in response to the question "Is there a difference between Ella and Plan B?" Please see Results, Comment #2 for additional methods clarifications
- Line 107 (there is an additional period there).
- Line 107 "In 40% of the calls among those who expressed no familiarity"--is "no familiarity" different than "lack of familiarity"? From the reported numbers, it doesn't appear to be different so I would recommend using the same terminology.
- Line 108: Authors mention the calls being transferred, but don't mention that in the methods as being a data collection point
- Consider adding a flow diagram of the calls and how many had ECPs available vs. not and the flow through the questions based on the caller? if not for all the script questions, for the ones in particular that were analyzed for this sub-study
Discussion
- Authors mention that only 3% of pharmacies had UPA available, but in Methods they said that pharmacies that didn't have ECPs were excluded--so are these results only 3% of the original sample? (Line 158)
- Methods of coding answers as dichotomously correct or incorrect and then being checked by another researcher should be added to the methods (Line 166)
- Line 198 "f in efficacy" is a typo?
- Overall, the discussion is mainly focused on limitations to the study and not a larger discussion of previous research in this area and how this contributes to the overall larger body of research and what this means for next steps. The authors should augment the discussion with some of these pieces to make it an overall stronger manuscript.
Author Response
Intro
- UPA--it would be helpful to mention that UPA's effectiveness doesn't diminish over time (as ECP's does).
The text now reads lines 35-36 “UPA is recommended for use within 120 hours of unprotected sexual intercourse and is more effective at preventing ovulation and pregnancy when compared to levonorgestrel ECPs (LNG ECPs) at 1, 3, and 5 days [1].
- Line 51-suggest stating "government-issued identification" instead of "ID"
We have adopted this suggested change. Lines 51-56 now read “A 2013 nationwide mystery client survey with callers posing as 17-year-old women seeking LNG ECPs also exposed additional barriers to access, with pharmacy staff incorrectly indicating that the medication required a prescription, could not be bought by minors or by male customers, stating that parental notification was required, and that the purchaser needed to present valid government-issued identification [6].”
- The authors make the argument that overall knowledge of UPA is low (amongst health care providers) and availability--can they clarify their hypothesis for this study to examine pharmacy staff knowledge?
We have clarified the hypothesis. The text now reads lines 56-61 “Currently there is no published data describing the information provided by pharmacy staff regarding UPA. We conducted a secondary analysis of a previously published mystery client telephone survey that assessed statewide availability of UPA and LNG ECPs in Hawai’i that included verbatim responses from pharmacy personnel regarding the differences between UPA and LNG ECPs. Our outcomes were to categorize the type of information voluntarily provided over the phone and to assess the accuracy of these responses.”
Methods
- The table of "correct" answers lists LNG as being effective for 72hr time-frame, can the authors clarify if the 5-day window that is evidence-based was also acceptable?
In our study we used the FDA labeling for our “correct” answer as we believed that pharmacy staffers answering the telephone would be more likely to use industry standard labels as their primary source of information.
We have also added to the methods section more description of the coding process. Lines 113 to 119 now read: “For this secondary analysis, we analyzed verbatim answers to the question “Is there a difference between ella® and Plan B?” (Figure 1) and any additional related responses that were documented by the researchers during the course of the semi-structured interview. Two authors (GK, MT) separately conducted content analysis, which allows for distillation of responses into overarching categories and response counting. Discrepancies in coding were discussed to reach consensus in classification of category. Each author classified answers as correct or incorrect information using FDA fact sheets. Discrepancies were again discussed to reach agreement [7, 8]. “
- It would be helpful if authors could help map the outcomes reported in Results and Table 2 onto the questions that were asked during the script and how answers were characterized. Table 1 helps talk about what was considered "correct", but then they report "lack of familiarity" and therefore it isn't clear how/why calls were characterized that way. Table 1 and Table 2 could list categories (Efficacy, Mechanism, etc) in the same order
We have included a new Figure 1 to better demonstrate the scripted questions and the scope of answers considered to show a “lack of familiarity.”
We have reorganized Table 2 to match the order of Table 1.
Category |
Frequency of Each Category Addressed |
Frequency of Correct Responses |
|||
Lack of Familiarity with UPA |
39% |
(141/358)
|
-- |
|
|
Timing of Drug Administration |
39% |
(139/358) |
18% |
(25/139) |
|
Prescription Requirement |
12% |
(41/358) |
71% |
(29/41) |
|
Drug Type |
55% |
(197/358) |
29% |
(58/197) |
|
Efficacy |
12% |
(44/358) |
25% |
(11/44) |
|
Results
- Can the authors clarify how there were 198 pharmacies called-but only 192 were mystery caller patients and 166 were physicians? Was every pharmacy not called by both the patient and physician? If not, can they clarify in methods?
We have better clarified the methods section better describe these perceived discrepancies in numbers of calls made. The methods section lines 63-84 now reads:
“We performed a secondary analysis of an observational population-based dataset assessing statewide availability of ECPs in retail pharmacies [2]. Two online phone books were cross-referenced with pharmacy chain websites to generate a list of all 198 unique retail pharmacies in Hawaiʻi. Pharmacies were excluded if the phone number was found to be disconnected or incorrect or if research staff was unable to reach the pharmacy after 3 attempted calls. Additional exclusions included pharmacies unwilling to provide information over the telephone and incomplete data collection at the time of call, Additionally, pharmacies that reported not carrying ECP, which did not provide information to patients over the phone, and specialty pharmacies were excluded. This study was reviewed by University of Hawaiʻi Institutional Review Board which deemed it to be “not human subjects research.”
Trained research assistants posing as 18-year-old patients attempting to fill a UPA prescription made calls to pharmacies utilizing a semi-structured questionnaire asking about the pharmacy’s ability to fill a prescription for UPA. They also asked about the availability of other types of ECPs and concluded with an open-ended question about the difference between UPA and LNG ECPs A physician member of the research team also made calls to pharmacies utilizing a similar semi-structured questionnaire, attempting to identify pharmacies where their patient could fill a UPA prescription, if additional LNG ECPs were available, but only asked the open-ended question about the difference between UPA and LNG ECPs if the pharmacy carried any ECPs. After pilot testing amongst departmental faculty and beta testing with 10 pharmacies in North Carolina, having the physician caller solicit more information about medications not on formulary was deemed inauthentic to reality, whereas for the patient caller soliciting information was deemed authentic to a real world patient interaction.”
We further added later in the methods section lines 104- 108 now reads: “The callers documented the pharmacy personnel responses to all questions, including additional remarks made throughout the call. Calls were made Monday through Saturday, 8 am through 8 pm, from December 2013 to July 2014. Three unique attempts were made to each pharmacy by the physician and patient mystery clients, varying the time of day and the day of the week. Calls were staggered by 4 weeks between the physician and patient mystery clients. “
- The outcome of "lack of familiarity with UPA"--was that in response to the question "Is there a difference between Ella and Plan B?" Please see Results, Comment #2 for additional methods clarifications
Thank you for this suggestion. We have created a new Figure 1 to better explain the range of answers that were included in the “lack of familiarity with UPA” answer
- Line 107 (there is an additional period there). This has been corrected.
- Line 107 "In 40% of the calls among those who expressed no familiarity"--is "no familiarity" different than "lack of familiarity"? From the reported numbers, it doesn't appear to be different so I would recommend using the same terminology.
Thank you for this suggestion. Lines 148- 150 now read: In 40% (56/141) of the calls among those who expressed a lack of familiarity with UPA, the initial respondent transferred the call to a pharmacist or more knowledgeable colleague.
- Line 108: Authors mention the calls being transferred, but don't mention that in the methods as being a data collection point.
We have clarified that answers were recorded verbatim and now also include this into our new Figure 1. Lines 113-119 reads: “For this secondary analysis, we analyzed verbatim answers to the question “Is there a difference between ella® and Plan B?” (Figure 1) and any additional related responses that were documented by the researchers during the course of the semi-structured interview. Two authors (GK, MT) separately conducted content analysis, which allows for distillation of responses into overarching categories and response counting. Discrepancies in coding were discussed to reach consensus in classification of category. Each author classified answers as correct or incorrect information using FDA fact sheets. Discrepancies were again discussed to reach agreement [7, 8]. “
- Consider adding a flow diagram of the calls and how many had ECPs available vs. not and the flow through the questions based on the caller? if not for all the script questions, for the ones in particular that were analyzed for this sub-study
Our new Figure 1 better maps the mystery caller’s flow through the questions. We did not include in our results the numbers of pharmacies with UPA vs. LNG ECPs as this information was previously published.
Discussion
- Authors mention that only 3% of pharmacies had UPA available, but in Methods they said that pharmacies that didn't have ECPs were excluded--so are these results only 3% of the original sample? (Line 158)
We have sought to further clarify the exclusions in the text and in new Figure 2. It is true that all mystery caller patient’s asked about the difference between UPA and LNG ECPs as this was deemed by the research team to be a real-life likelihood during the beta testing of the script with pharmacies. With the mystery caller physician calls, asking about the difference between two medications when neither was in stock was not successfully in the beta testing phase of the questionnaire, and this question was dropped. A total of 16 pharmacies carried neither ECP type that the mystery caller physician contacted. Lines 73-84 now read:
Trained research assistants posing as 18-year-old patients attempting to fill a UPA prescription made calls to pharmacies utilizing a semi-structured questionnaire asking about the pharmacy’s ability to fill a prescription for UPA. They also asked about the availability of other types of ECPs and concluded with an open-ended question about the difference between UPA and LNG ECPs A physician member of the research team also made calls to pharmacies utilizing a similar semi-structured questionnaire, attempting to identify pharmacies where their patient could fill a UPA prescription, if additional LNG ECPs were available, but only asked the open-ended question about the difference between UPA and LNG ECPs if the pharmacy carried any ECPs. After pilot testing amongst departmental faculty and beta testing with 10 pharmacies in North Carolina, having the physician caller solicit more information about medications not on formulary was deemed inauthentic to reality, whereas for the patient caller soliciting information was deemed authentic to a real world patient interaction.
- Methods of coding answers as dichotomously correct or incorrect and then being checked by another researcher should be added to the methods (Line 166)
Thank you for this suggestion. The methods section lines 113-119 now reads: “For this secondary analysis, we analyzed verbatim answers to the question “Is there a difference between ella® and Plan B?” (Figure 1) and any additional related responses that were documented by the researchers during the course of the semi-structured interview. Two authors (GK, MT) separately conducted content analysis, which allows for distillation of responses into overarching categories and response counting. Discrepancies in coding were discussed to reach consensus in classification of category. Each author classified answers as correct or incorrect information using FDA fact sheets. Discrepancies were again discussed to reach agreement [7, 8].
- Line 198 "f in efficacy" is a typo? This typo has been corrected.
- Overall, the discussion is mainly focused on limitations to the study and not a larger discussion of previous research in this area and how this contributes to the overall larger body of research and what this means for next steps. The authors should augment the discussion with some of these pieces to make it an overall stronger manuscript.
We have added an additional citation to give context to the import of our study as filling a gap in the literature. We would be open to further refining the discussion if more guidance can be given on the direction to fill the needs of this EC specific journal addition. The first paragraph lines 197-206 now reads: “We examined the accuracy of information provided by retail pharmacy personnel when asked to describe differences between UPA and LNG during a call inquiring about a prescription for UPA. Callers seeking such information from retail pharmacies in Hawaiʻi received accurate information from a small percentage of respondents. Lack of familiarity with UPA was a common theme encountered. This could be linked to the limited availability of UPA in pharmacies, as only 3% of retail pharmacies reported same day availability and 25% reported the ability to order UPA in the primary analysis of this data [2]. To date there has been no published literature on information that callers receive over the phone from pharmacies regarding UPA, but there is a large body of literature citing misinformation from pharmacy staff to callers for LNG ECPs. As the majority of patients access ECPs from chain pharmacies or “big box” stores, we sought to address this gap in the literature [11].”

Reviewer 2 Report
This is an interesting study, as it shows that information given about different products for emergency contraception needs to be improved in the area of community pharmacy.
Some things I would like to comment:
- Introduction: When saying that UPA access is limited, in what sense is this said? Because there is a need of a prescription? This is related with what is said in line158 of discussion related to the limited availability of UPA.
- Line 198, I think that there is a typo mistake “ f in efficacy”
- In the manuscript, there is no mention about possible adverse effects, at least the most common. Did not the authors took this into account?
Author Response
Reviewer 2:
Comments and Suggestions for Authors
This is an interesting study, as it shows that information given about different products for emergency contraception needs to be improved in the area of community pharmacy.
Some things I would like to comment:
- Introduction: When saying that UPA access is limited, in what sense is this said? Because there is a need of a prescription? This is a secondary analysis of open-ended questions from the original study showing 3% of retail pharmacies in the state of Hawai’i carried UPA. We have further described the lack of availability of UPA in the introduction. Lines 38-39 now read: “Access to UPA through retail pharmacies is limited when compared to LNG ECPs as few pharmacies carry the medication [2].”
- Line 198, I think that there is a typo mistake “ f in efficacy” This has been corrected.
- In the manuscript, there is no mention about possible adverse effects, at least the most common. Did not the authors took this into account?
- In this study, we sough to analyze the information provided by pharmacy staff when prompted with a very general question “What is the difference between ella® and Plan B?” Of all the verbatim answers given, none were related to side effects of the medications. We therefore did not include it into our paper.
